# Nursing students' self-efficacy in lifestyle counselling: Associations with learning methods

Sara Alenius[1,2,3]*, Marie Rask[1,2,3], Albert Westergren[1,2,3], Petra Nilsson Lindström[1,3], Marie Nilsson[1,3], Lina Behm[1,2,3]

**1** Faculty of Health Sciences, Department of Nursing and Integrated Health Sciences, Kristianstad University, Kristianstad, Sweden, **2** The PRO-CARE Group, Faculty of Health Sciences, Kristianstad University, Kristianstad, Sweden, **3** Centre for Food Health and Retail at Kristianstad University (FOHRK), Kristianstad University, Kristianstad, Sweden

* Sara.alenius@hkr.se

## Abstract

Several non-communicable diseases are strongly linked to lifestyle factors, making preventive measures essential. One effective approach is lifestyle counselling, which has demonstrated promising results in the prevention, treatment, and management of these diseases. However, despite its potential, patients often do not receive lifestyle counselling to the extent required. The existing literature indicates that one contributing factor is low self-efficacy in lifestyle counselling among nurses. This study aimed to explore nursing students' self-efficacy in lifestyle counselling and its association with self-assessed learning methods. Nursing students (n = 310) completed a questionnaire that had a cross-sectional study design at a university in southern Sweden. The self-efficacy in lifestyle counselling scale (SELC20+20) was used to measure self-efficacy in lifestyle counselling, and additional questions about learning methods were included to assess Bandura's sources of self-efficacy. Multiple linear regression was used to explore the relationship between self-efficacy in lifestyle counselling and self-assessed learning methods. The mean total knowledge score was significantly higher than that of the mean total ability score. The learning methods that were significantly associated with self-efficacy in knowledge of lifestyle counselling after adjustment for age and sex, were: own search for knowledge, theoretical knowledge through education, receiving feedback on counselling, personal experiences, observed lifestyle counselling, and experience in lifestyle counselling ($R^2 = 0.30$). Learning methods that were significantly associated with self-efficacy in lifestyle counselling ability, after adjusting for age and sex, were: own search for knowledge, personal experiences, theoretical knowledge through education, and receiving feedback on counselling ($R^2 = 0.33$). The results indicated that mastery experiences, vicarious experiences, and social persuasion were significantly associated with knowledge of lifestyle counselling, while mastery experiences and social persuasion were significantly associated with lifestyle counselling ability. Further research

**Data availability statement:** Data are available from the Swedish National Data Service (https://doi.org/10.5878/pama-e009), and upon request from the first author.

**Funding:** This study was supported by Kristianstad University. The funder had no role in study design, data collection and analysis, decision to publish, or preparation of the manuscript.

**Competing interests:** The authors have declared that no competing interests exist.

**Abbreviations:** NCD, Non-communicable disease; WHO, World Health Organization; ECTS, European Credit Transfer and Accumulation System; SELC20+20, Self-efficacy in lifestyle counselling scale; B, Regression coefficient.

is needed to deepen our understanding of how self-efficacy in lifestyle counselling develops among nursing students.

## Introduction

Non-communicable diseases (NCDs) are the leading causes of morbidity and mortality worldwide [1]. Conditions such as heart disease, diabetes, stroke, and cancer are strongly linked to lifestyle factors, including poor diet, tobacco use, harmful alcohol consumption, and physical inactivity [2]. Existing literature shows that lifestyle counselling is a cost-effective way to prevent NCDs, reduce premature mortality, and promote health equity [3–5]. The World Health Organization (WHO) recommends healthy-lifestyle counselling in primary care as an essential intervention for tackling NCDs [6,7]. In Sweden, lifestyle counselling in health care is an important part of the fight against NCDs [8]; according to the Swedish Health and Medical Services Act, all health care professionals are required to counsel patients on lifestyle habits [9,10]. National guidelines for managing unhealthy lifestyle habits emphasize screening and structured counselling at three levels: basic advice, advisory counselling, and qualified counselling in the following areas: tobacco and nicotine use, alcohol consumption, physical activity, and eating habits [10,11]. Despite the recognised importance of lifestyle counselling, medical records from primary health care show that only approximately 12% of patients had any form of lifestyle counselling documented in their patient journal during the year 2022 [12]. Research suggests that several barriers for nurses in providing lifestyle counselling are rooted in low self-efficacy in lifestyle counselling [13,14].

According to Bandura, self-efficacy refers to context-specific confidence in one's ability to perform the actions necessary to achieve the desired outcomes [15]. Self-efficacy depends on a person's subjective assessment of their own capability within a specific context and operates independently of their objective level of competence [16]. High self-efficacy can enhance effort invested in a task and resilience to setbacks [17]. Bandura has identified four primary sources of self-efficacy: mastery experiences, vicarious experiences, social persuasion, and physiological and emotional states [15]. Self-efficacy is primarily developed through mastery experiences–instances of success in performing a task, such as conducting a lifestyle counselling session [15]. Observing others succeed at a similar task, known as vicarious experience, also strengthens self-efficacy, especially observing someone similar to oneself when one is uncertain about one's own abilities [15]. Social persuasion, in the form of supportive and constructive feedback from significant individuals, such as teachers or peers, can foster self-efficacy, while negative criticism can undermine it [18]. In addition, individuals often interpret their physiological and emotional states as indicators of their capabilities [15]. For example, overwhelming anxiety or nervousness during a lifestyle counselling session may be perceived as a lack of competence, potentially creating a negative feedback loop that can diminish self-efficacy in counselling [19].

To acquire fundamental self-efficacy in lifestyle counselling during nursing education, students do not require only knowledge and understanding of lifestyle habits. They also need the ability, i.e., the skills to apply their knowledge practically, analyse the situation, and evaluate how they need to adjust their counselling to the person in front of them. Bloom's taxonomy is a framework for classifying the stages of learning into six levels, progressing from a lower order to higher order with respect to cognitive complexity: remember, understand, apply, analyse, evaluate, and create. The framework also divides knowledge into four hierarchical levels: factual, conceptual, procedural, and metacognitive. This taxonomy was originally developed by Bloom in 1956 and later revised by Anderson and Krathwohl in 2001 [20]. Nursing is a practical profession in which remembering and understanding knowledge can largely be developed in a classroom setting on campus, whereas practical skills and abilities for lifestyle counselling, which reach learning levels such as applying, analysing, and evaluating, primarily need to be developed through practice, preferably in clinical training [21]. Research on how various learning methods influence the development of self-efficacy in lifestyle counselling remains limited, with most studies being outdated, emphasising the need for more contemporary investigations. Previous research has suggested that learning experiences that integrate multiple sources of self-efficacy are closely linked to higher levels of self-efficacy [22]. Their research demonstrated that combining theoretical knowledge, such as delivering risk information for patient education, with hands-on practical experience was particularly effective in enhancing students' confidence in conducting lifestyle counselling. Laschinger et al. [23] explored the relationship between learning methods and self-efficacy in lifestyle counselling among nursing and medical students [23]. Their results indicated that nursing students' self-efficacy was most strongly associated with learning-specific health promotion strategies in the classroom and through practical applications. In contrast, medical students benefitted most from practice opportunities, performance feedback, and role modelling; this highlights discipline-specific variations in how self-efficacy can be effectively developed [23].

Given the limited research in this area and the clinical need, it is essential to explore how different learning methods influence self-efficacy in lifestyle counselling. The insights gained can contribute to the identification of effective educational strategies that strengthen students' confidence and competence in this crucial aspect of health care practice. Therefore, this study aimed to explore nursing students' self-efficacy in lifestyle counselling and its association with self-assessed learning methods.

## Methods

### Study design, context, and participants

This study used a cross-sectional design. Survey data were collected from nursing students at a university in southern Sweden in January 2023. Nursing education in Sweden consists of three years of full-time studies (180 ECTS) of mainly nursing; however, topics also include medicine and public health [24]. During the three years, there are five periods of clinical training, consisting of a total of 50 ETCS. Simulation exercises are also used for clinical practice [25]. Two researchers (SA and LB) visited teaching sessions during semesters 2–6 of the nursing education at a university in southern Sweden. SA provided oral information about the study and its voluntary nature; then, the questionnaires were distributed in the classroom, and the students were given time to answer the questions. A total of 347 students were present and 310 (89%) students chose to participate.

### Ethical considerations

This study was approved by the local Health Sciences Ethics Council (dnr U2022-2.1.12–2112). The study was conducted in accordance with the Declaration of Helsinki and Swedish Law of Research Ethics [26,27]. The questionnaire did not contain any personal information, and to hand in the questionnaire was to give informed consent to participate in the study. The data were collected in a higher education environment, which may have compelled the students to answer. To reduce this risk, autonomy was emphasised in the oral information that students were provided about the study.

Furthermore, if the students did not want to participate, they were asked to submit a blank questionnaire to the researchers to ensure their anonymity.

## Instrument

**The self-efficacy in lifestyle counselling scale (SELC 20 + 20).** This study was preceded by a study which developed and qualitatively assessed an instrument to measure self-efficacy in lifestyle counselling (SELC 20 + 20) [28]. The instrument includes two separate constructs, knowledge and ability, with 20 items each for self-assessed self-efficacy in lifestyle counselling. Each construct (knowledge and ability) comprises four parts (tobacco/nicotine use, alcohol consumption, physical activity, and eating habits). The instrument is answered using a 4-point Likert-scale ranging from "very insecure" to "very sure". All items of the SELC 20 + 20 are ordinal, ranging from 0–3 points per item, with each part scoring 0–15 points. Each construct (knowledge and ability) can be separately summed to obtain a total score of 0–60 points, resulting in two separate total scores. All items with response category endorsements, rigid validity, and reliability evaluations of the instrument were presented by Alenius et al. [28].

**Learning methods.** Additional single items regarding the following learning methods were used in the present study: own search for knowledge, personal experiences, theoretical knowledge through education, observed lifestyle counselling, experience in lifestyle counselling, and receiving feedback on counselling. The items were answered using a 4-point Likert scale ranging from 0 to 3, "not at all" to "to a very high degree". A 4-point Likert scale was used to maintain consistency with the SELC 20 + 20 instrument to minimize cognitive burden for the respondents. The items: experience in lifestyle counselling, observed lifestyle counselling, and receiving feedback on counselling were developed according to Bandura's sources of self-efficacy [15] (see Table 1). The items: own search for knowledge and theoretical knowledge through education were developed according to Bloom's taxonomy [20] (see Table 1).

## Data analysis

Data analysis consisted of one descriptive segment and one analytical segment. Due to the qualitative nature of the items, linearised (through Rasch model analysis) scores of self-efficacy in knowledge and ability were used throughout the analyses [28]. The data were initially descriptively analysed as frequencies (n) and percentages (%). In the analytical segment, the data were checked for underlying assumptions for each analysis. The excluded participants were compared with the included participants using the chi-square test. A paired-sample t-test was used to compare the means of knowledge and ability. Independent samples t-test was used for bivariate analyses of associations between the dependent variable of self-efficacy for lifestyle counselling in each construct, and the following independent variables; age divided according to median (≤25 = 0, >26 = 1), sex (women = 0, men = 1), educational level (upper secondary school = 0, university = 1) and previous health care education (yes = 0, no = 1). A one-way between-group ANOVA was

**Table 1. Theoretical source for learning method items.**

| Source | Item |
| --- | --- |
| Factual-, conceptual- and procedural knowledge[a] | Own search for knowledge |
| | Theoretical knowledge through education |
| Mastery experiences[b] | Personal experiences |
| | Experience in lifestyle counselling |
| Vicarious experiences[b] | Observed lifestyle counselling |
| Social persuasion[b] | Receiving feedback on counselling |

[a]Bloom's revised taxonomy;

[b]Self-efficacy theory.

used for the bivariate analysis of associations between the dependent variable of self-efficacy in lifestyle counselling in each construct and the following independent variables: semester [2–6], own search for knowledge (0–3), personal experiences (0–3), theoretical knowledge through education (0–3), observed lifestyle counselling (0–3), experience in lifestyle counselling (0–3) and receiving feedback on counselling (0–3). Multiple linear regression was then used to examine the associations between the dependent variable, self-efficacy for lifestyle counselling in knowledge and ability, and the independent variables of learning methods (own search for knowledge, personal experiences, theoretical knowledge through education, observed lifestyle counselling, experience in lifestyle counselling, and receiving feedback on counselling), adjusted for age (continuous variable) and sex. Dummy variables for each response category were created for the independent variables of the learning methods. The response category "To a very high degree" was used as the reference category, compared to "Not at all", "To a low degree", and "To a high degree". In the knowledge construct, the response categories "Not at all" and "To a low degree" were merged into one dummy variable because of low response rate, "Not at all" ($n = 2$). In both the knowledge and ability constructs, the tolerance (threshold >0.25) and VIF (threshold <5) values were good. Knowledge self-efficacy was examined using three regression models. In the first model (method: enter) and the second model (method: backward), all the independent variables listed above were included. In the final model, the significant independent variables from the backward model were included and adjusted for age and sex. Self-efficacy in ability was similarly explored using three regression models (method: enter, method: backward, and method: enter). The initial models similarly included all independent variables listed above, and the final model included significant independent variables from the backward model. IBM SPSS software was used for all statistical analyses, and the alpha level of significance was set at $p < 0.05$ [29].

## Results

Among the 310 participating students, 257 (83%) completed all 56 items (20 + 20 SELC items, 6 + 6 items on learning methods, and 4 demographic questions). Only complete cases were included in the analysis ($n = 257$). The excluded participants did not differ in terms of sex ($p = 0.783$), age ($p = 0.204$), educational level ($p = 0.461$), or previous health care education ($p = 0.853$) compared to those included. Among the included participants, the median age was 25 (Q1-Q3, 23–31; range, 20–56), 89.1% ($n = 229$) were women, 12.5% ($n = 32$) had a previous university degree, and 38.1% ($n = 98$) had previous health care education. Regarding self-efficacy in knowledge for lifestyle counselling, the mean total score was 37.22 (SD 6.77; range, 16–60) of which 1.2% ($n = 3$) scored the maximum. Considering self-efficacy in ability for lifestyle counselling, the mean total score was 35.91 (SD 7.30; range, 12–60) of which 0.4% ($n = 1$) scored the maximum. The mean total score in knowledge was significantly higher than the mean total score in ability (see S1 Table). Bivariate analysis showed that those aged 26 years or older scored significantly higher than younger participants and men scored significantly higher than women on self-efficacy in lifestyle counselling ability. There were no significant differences in educational level, previous health care education, or among the different semesters (see S1 Table). There were significant differences in the total scores for both the knowledge and ability constructs between the response categories of all learning method items ($p < 0.001$) (see S2 Table).

Controlling for age and sex, the multiple linear regression with self-efficacy in knowledge of lifestyle counselling as the dependent variable showed significant associations with the following independent variables (in descending order) (B): own search for knowledge (−5.80- −2.12), theoretical knowledge through education (−4.55--2.63), receiving feedback on counselling (−4.20), personal experiences (−2.93), observed lifestyle counselling (−2.15--2.12), and experience in lifestyle counselling (−1.88) (see Table 2). The significant model explained 30% of the variance in self-efficacy in knowledge of lifestyle counselling among nursing students.

Controlling for age and sex, the multiple linear regression with self-efficacy in ability for lifestyle counselling as the dependent variable showed significant associations with the following independent variables (in descending order) (B): own search for knowledge (−8.00--2.77), personal experiences (−7.39--3.12), theoretical knowledge through education

**Table 2. Multiple linear regression with self-efficacy in knowledge for lifestyle counselling (SELC20 + 20 linearised total score) as the dependent variable, *n* = 257. Method = Enter.**

| Significant independent variables | B | SE | 95% CI | β | t | p-value |
|---|---|---|---|---|---|---|
| (Constant) | 43.50 | 1.80 | 39.95-47.06 | | 24.10 | < 0.001 |
| Age | 0.02 | 0.05 | −0.07-0.12 | 0.03 | 0.47 | 0.638 |
| Sex | 1.86 | 1.16 | −0.43-4.15 | 0.09 | 1.60 | 0.110 |
| **Own search for knowledge**[a] | | | | | | |
| Not at all and To a low degree[b] | −5.80 | 1.11 | −7.99 to −3.61 | −0.32 | −5.21 | **< 0.001** |
| To a high degree | −2.12 | 0.87 | −3.84 to −0.41 | −0.16 | −2.44 | **0.016** |
| **Theoretical knowledge through education**[a] | | | | | | |
| To a low degree | −4.55 | 1.10 | −6.72 to −2.37 | −0.25 | −4.12 | **< 0.001** |
| To a high degree | −2.63 | 0.86 | −4.32 to −0.98 | −0.19 | −3.06 | **0.002** |
| **Receiving feedback on counselling**[a] | | | | | | |
| Not at all | −4.20 | 1.14 | −6.46 to −1.95 | −0.21 | −3.67 | **< 0.001** |
| **Personal experiences**[a] | | | | | | |
| To a low degree | −2.93 | 0.97 | −4.84 to −1.03 | −0.17 | −3.03 | **0.003** |
| **Observed lifestyle counselling**[a] | | | | | | |
| To a high degree | −2.15 | 0.87 | −3.85 to −0.44 | −0.16 | −2.48 | **0.014** |
| To a low degree | −2.12 | 0.98 | −4.04 to −0.19 | −0.14 | −2.17 | **0.031** |
| **Experience in lifestyle counselling**[a] | | | | | | |
| To a low degree | −1.88 | 0.85 | −3.55 to −0.20 | −0.12 | −2.20 | **0.029** |

[a]Reference category was "To a very high degree";

[b]Response categories merged into one dummy variable because of low response rate on Not at all (n = 2); B: regression coefficient; SE: standard error; CI: confidence interval; β: standardized regression coefficient; Durbin-Watson = 2.10; *F* = 10.96; Model *p*-value < 0.001; Adjusted $R^2$ = 0.30. There were no signs of multicollinearity (tolerance: 0.7). Bolded p-values indicate statistical significance (p < 0.05) for independent variables.

(−5.81--3.38), and receiving feedback on counselling (−3.41) (see Table 3). The significant model explained 33% of the variance in self-efficacy in ability for lifestyle counselling among nursing students.

## Discussion

This study aimed to explore nursing students' self-efficacy in lifestyle counselling and its association with self-assessed learning methods. Understanding how students develop self-efficacy in terms of both the knowledge and practical skills required for lifestyle counselling is essential for optimising educational strategies. The results showed that the mean total score on knowledge of lifestyle counselling was significantly higher than the mean total score on the ability to provide lifestyle counselling (see S1 Table). This aligns with a previous study by Alenius et al. [28], which suggests that self-efficacy in knowledge serves as a prerequisite for self-efficacy in ability. According to Bloom's taxonomy [20], remembering and understanding knowledge are positioned at a lower level than the practical ability to apply knowledge, such as applying, analysing, and evaluating [20]. This suggests that knowledge should be easier to achieve than ability [21]. According to Alenius et al. [28], the constructs of knowledge and ability are closely interconnected, reinforcing the need to integrate learning approaches that address both dimensions simultaneously [28].

Furthermore, the results showed that regarding the knowledge construct, there were no significant differences in total scores among the different age groups, sexes, educational levels, or among those with or without previous health care education. However, within the ability construct, the older age group (>26 years) and men had significantly higher total scores (see S1 Table). Surprisingly, there were no significant differences between the semesters of nursing education, either in their knowledge or ability. In contrast, progression towards higher self-efficacy after progressively more education

**Table 3. Multiple linear regression with self-efficacy in ability for lifestyle counselling (SELC20+20 linearised total score) as the dependent variable, *n*=257. Method=Enter.**

| Significant independent variables | B | SE | 95% CI | β | t | p-value |
|---|---|---|---|---|---|---|
| (Constant) | 41.00 | 1.76 | 37.54-44.46 | | 23.35 | < 0.001 |
| Age | 0.05 | 0.05 | −0.05-0.15 | 0.05 | 0.98 | 0.328 |
| Sex | 2.42 | 1.22 | 0.02-4.83 | 0.10 | 1.99 | **0.048** |
| **Own search for knowledge**[a] | | | | | | |
| Not at all | −8.00 | 2.79 | −13.50 to −2.51 | −0.17 | −2.87 | **0.004** |
| To a low degree | −2.77 | 1.02 | −4.78 to −0.75 | −0.15 | −2.71 | **0.007** |
| **Personal experiences**[a] | | | | | | |
| Not at all | −7.39 | 2.17 | −11.66 to −3.12 | −0.20 | −3.41 | **0.001** |
| To a low degree | −5.07 | 1.15 | −7.34 to −2.79 | −0.28 | −4.39 | **< 0.001** |
| To a high degree | −3.12 | 0.90 | −4.89 to −1.34 | −0.21 | −3.46 | **0.001** |
| **Theoretical knowledge through education**[a] | | | | | | |
| To a low degree | −5.81 | 1.18 | −8.15 to −3.48 | −0.31 | −4.91 | **< 0.001** |
| To a high degree | −3.38 | 0.90 | −5.15 to −1.61 | −0.23 | −3.76 | **< 0.001** |
| **Receiving feedback on counselling**[a] | | | | | | |
| Not at all | −3.41 | 1.19 | −5.76 to −1.06 | −0.16 | −2.86 | **0.005** |

[a]Reference category was "To a very high degree". B: regression coefficient; SE: standard error; CI: confidence interval; β: standardized regression coefficient. Durbin-Watson=2.24; $F$=13.39; Model *p*-value<0.001; Adjusted $R^2$=0.33. There were no signs of multicollinearity (tolerance: 0.7). Bolded p-values indicate statistical significance (p<0.05) for independent variables.

and clinical experience would have been logical. One explanation might be the lack of a structured educational component, with recurring training in lifestyle counselling, throughout the nursing curriculum. Another explanation could be the low frequency of lifestyle counselling in health care in Sweden, which might lead to students not getting the opportunity to practice during their clinical placements. However, the lack of significant differences between the semesters might also be caused by the small number of participants representing each semester in the present study. The instrument has not yet been used in any study population other than nursing students, such as clinical nurses, which makes it difficult to distinguish between high and low total scores. A few individuals (n=4) self-assessed maximum total scores, which might have a negative impact on their receptivity to learning. Although, according to Schunk et al. [16], having a slightly higher level of self-efficacy than one's objective level of knowledge or ability can facilitate learning, because it can lead to a higher effort being put into the task [16].

Moreover, for knowledge, the results indicated that all learning methods, representing both the factual, conceptual, and procedural levels of knowledge in Bloom's taxonomy [20], as well as three of Bandura's sources of self-efficacy (mastery experience, vicarious experience, and social persuasion) [15], were significantly associated with self-efficacy in knowledge for lifestyle counselling (see Table 2). Regarding ability, a somewhat surprising result was that only two of Bandura's sources of self-efficacy (mastery experience and social persuasion) [15], along with the first three levels of knowledge in Bloom's taxonomy [21], were significantly associated with self-efficacy in lifestyle counselling (see Table 3). The results showed that personal experiences in both constructs and lifestyle counselling experience in the knowledge construct, representing mastery experiences, had significant associations with self-efficacy in both knowledge and ability for lifestyle counselling. Similarly, a study by Malatskey et al. [30] found in focus group discussions that experience-based learning through a course in lifestyle counselling, including theoretical knowledge and six sessions of lifestyle counselling per student, enhanced medical students' confidence and competence in lifestyle counselling [30]. In the present study, receiving feedback on counselling, representing social persuasion, also had significant associations with self-efficacy in terms of both knowledge of and ability for lifestyle counselling. This result aligns with a study by Rowbotham et al. [31] examining

the effect of clinical nursing instructors on students' self-efficacy during practical clinical experiences and identified several instructor behaviours that positively influenced self-efficacy [31]. Students with high self-efficacy reported that instructors who provided constructive feedback, highlighted their strengths and weaknesses, frequently observed their performance, clearly communicated expectations, offered positive reinforcement, and corrected mistakes without belittling them, significantly enhanced their confidence [31]. Thus, the results indicate the importance of reflective practice and guided feedback in developing confidence and competence in lifestyle counselling. A surprising result of the present study was that vicarious experiences did not have a significant impact on the ability to provide lifestyle counselling. According to Bandura, vicarious experiences are an important source of self-efficacy because observing others similar to oneself when they succeed at the task can enhance self-efficacy beliefs [15]. Contrary to our results, a qualitative study by Zamani-Alavijeh et al. [32] found that vicarious experiences positively affected clinical health educators' self-efficacy in health education [32]. In this study, observing others enhanced self-efficacy in terms of theoretical knowledge. This means that observing lifestyle counselling and potentially seeing successful performances modelled by peers were significantly associated with higher self-efficacy in knowledge of lifestyle counselling. However, vicarious experiences were not significantly associated with self-efficacy in ability to provide lifestyle counselling. One possible explanation might be that vicarious experiences provide conceptual knowledge; that is, students gain an understanding of the process, strategies, and potential challenges. However, gaining self-efficacy in ability may require hands-on practice. Without actively engaging in counselling, students may lack self-efficacy in the procedural skills and confidence needed for their ability. Furthermore, knowledge primarily enhance cognitive learning and understanding of concepts, strategies, and techniques. Ability, on the other hand, involves psychomotor learning, the practical application of skills, which may require direct engagement and practice. Thus, our study suggests that achieving self-efficacy in lifestyle counselling requires practical training. Currently, such training is often lacking in nursing students' clinical practice. Consequently, there is a compelling need to develop and integrate targeted training opportunities to equip nurses with the requisite skills and confidence to deliver lifestyle counselling effectively in clinical environments. Preferably, training in lifestyle counselling should be recurring throughout the entire nursing curriculum, rather than being confined to isolated sessions.

Furthermore, because of the relatively low adjusted R-squared values, 30% for knowledge and 33% for ability (see Tables 2 and 3), additional factors influencing self-efficacy in lifestyle counselling appeared to be unaccounted for in this study. This gap highlights the need for further investigation, preferably by directly engaging students, for example, in focus group discussions, to better understand how they perceive and acquire the knowledge and skills necessary for lifestyle counselling. Exploring their experiences and perspectives could provide valuable insights into learning processes and uncover additional factors that contribute to the development of self-efficacy in this area. These results can inform the design of future targeted and effective educational interventions.

## Strengths and limitations

The strength of the internal validity was that the instrument used was psychometrically tested in a previous study [28]. The psychometric evaluation showed good measurement properties regarding reliability and that the instrument was well suited for group-level analyses, as used in the present study. A limitation of this study was that the instrument's stability over time had not been tested with test-retest. The items are ordinal in nature; hence, logistic regression should be well suited. However, it was not possible to set a cut-off point between low and high self-efficacy, because it is a continuum where a higher total score indicates a higher level of self-efficacy in lifestyle counselling. Fortunately, transformation to linearised scores had been performed in the psychometric study [28]. Therefore, multiple linear regression using linearised total scores for the knowledge and ability constructs was considered to perform data justice compared with multiple logistic regression. The strength of external validity was the high response rate (310 of 347 participants). However, data was collected through voluntary participation during classroom sessions, which poses the risk of response bias. All nursing students from semester 2–6, physically present at the university during data collection were invited to participate (n = 347).

Although the total number of nursing students, in the semesters included in the study, and registered at the university during the data collection, was 472. No attempt to reach students who were not physically present at the university were made, which was a limitation of the study. Other ways to reach students could have been to send the questionnaire by post or email. Nevertheless, there was a large range in total score both regarding self-assessed knowledge (16–60 p) and ability (12–60 p) for lifestyle counselling among the participants, which might indicate a variation in motivation and interest among the participants. Although 53 participants were excluded due to missing data, the missing data analysis showed no significant differences in age, sex, educational level, or previous health care education between those included and excluded. The generalisability of the result is limited because the sample consisted of students from a single nursing program at one university. To ensure the broader applicability of the results, future studies should include participants from multiple universities, countries and cultural contexts. Nonetheless, this study focused specifically on nursing students' self-efficacy in lifestyle counselling, an area with limited prior research, thereby making it exploratory.

## Conclusions

To effectively implement the structured levels of lifestyle counselling outlined in the national care program, it is crucial for nursing students to develop both theoretical knowledge of lifestyle factors and the practical skills and attitudes necessary for meaningful patient engagement during their education. The results indicated that mastery experiences, vicarious experiences, and social persuasion were significantly associated with self-efficacy in knowledge of lifestyle counselling, and that mastery experiences and social persuasion had significant associations with self-efficacy in lifestyle counselling ability. This implies the importance of practical training for the development of self-efficacy in lifestyle counselling. Future studies are needed to further investigate how self-efficacy in lifestyle counselling develops over time, preferably using both qualitative and longitudinal quantitative methods. By identifying and integrating the most effective learning methods, nursing education programs can not only improve students' self-efficacy but also contribute to aligning educational practice with national policy goals, ultimately strengthening the implementation of lifestyle counselling in clinical care.

## Supporting information

**S1 File.  SELC20 + 20.**
(PDF)

**S1 Table.  Bivariate associations of self-efficacy in knowledge and ability scores (*n* = 257).** Demographic questions.
(PDF)

**S2 Table.  Bivariate associations of self-efficacy in knowledge and ability scores (*n* = 257).** Learning methods.
(PDF)

## Acknowledgments

We would like to thank the nursing students who participated in this study.

## Author contributions

**Conceptualization:** Sara Alenius, Marie Rask, Albert Westergren, Petra Nilsson Lindström, Marie Nilsson, Lina Behm.

**Data curation:** Sara Alenius.

**Formal analysis:** Sara Alenius.

**Funding acquisition:** Marie Rask, Petra Nilsson Lindström, Marie Nilsson, Lina Behm.

**Investigation:** Sara Alenius, Lina Behm.

**Methodology:** Sara Alenius, Marie Rask, Albert Westergren, Lina Behm.

**Project administration:** Sara Alenius, Lina Behm.

**Supervision:** Albert Westergren, Lina Behm.

**Validation:** Albert Westergren, Lina Behm.

**Visualization:** Sara Alenius.

**Writing – original draft:** Sara Alenius, Albert Westergren.

**Writing – review & editing:** Marie Rask, Petra Nilsson Lindström, Marie Nilsson, Lina Behm.

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
