## [Decision Letter · Decision Letter 0]

26 Jun 2025

Dear Dr. Alenius,

Thank you for submitting your manuscript to PLOS ONE. After careful consideration, we feel that it has merit but does not fully meet PLOS ONE’s publication criteria as it currently stands. Therefore, we invite you to submit a revised version of the manuscript that addresses the points raised during the review process.

We look forward to receiving your revised manuscript.

Kind regards,

Pei Boon Ooi, Ph.D.

Academic Editor

PLOS ONE

**Journal Requirements:**

1. When submitting your revision, we need you to address these additional requirements. Please ensure that your manuscript meets PLOS ONE's style requirements, including those for file naming. The PLOS ONE style templates can be found at https://journals.plos.org/plosone/s/file?id=wjVg/PLOSOne_formatting_sample_main_body.pdf and https://journals.plos.org/plosone/s/file?id=ba62/PLOSOne_formatting_sample_title_authors_affiliations.pdf 2. Thank you for stating in your Funding Statement: LB(No grant numbers)Funder: Kristianstad Universityhttps://www.hkr.se/Statement:This study has been supported by Kristianstad University. The funder had no role in study design, data collection and analysis, decision to publish, or preparation of the manuscript.  Please provide an amended statement that declares *all* the funding or sources of support (whether external or internal to your organization) received during this study, as detailed online in our guide for authors at http://journals.plos.org/plosone/s/submit-now.  Please also include the statement “There was no additional external funding received for this study.” in your updated Funding Statement. Please include your amended Funding Statement within your cover letter. We will change the online submission form on your behalf. 3. We noted in your submission details that a portion of your manuscript may have been presented or published elsewhere. “This study used the same data as a previously published study (doi: 10.1186/s12955-024-02236-z, attached as a Related Manuscript). However, this study contains additional data about learning methods and had a completely different analysis than the previous study”.Please clarify whether this [conference proceeding or publication] was peer-reviewed and formally published. If this work was previously peer-reviewed and published, in the cover letter please provide the reason that this work does not constitute dual publication and should be included in the current manuscript. 4. We note that this data set consists of interview transcripts. Can you please confirm that all participants gave consent for interview transcript to be published? If they DID provide consent for these transcripts to be published, please also confirm that the transcripts do not contain any potentially identifying information (or let us know if the participants consented to having their personal details published and made publicly available). We consider the following details to be identifying information:- Names, nicknames, and initials- Age more specific than round numbers- GPS coordinates, physical addresses, IP addresses, email addresses- Information in small sample sizes (e.g. 40 students from X class in X year at X university)- Specific dates (e.g. visit dates, interview dates)- ID numbers Or, if the participants DID NOT provide consent for these transcripts to be published:- Provide a de-identified version of the data or excerpts of interview responses- Provide information regarding how these transcripts can be accessed by researchers who meet the criteria for access to confidential data, including:a) the grounds for restrictionb) the name of the ethics committee, Institutional Review Board, or third-party organization that is imposing sharing restrictions on the datac) a non-author, institutional point of contact that is able to field data access queries, in the interest of maintaining long-term data accessibility.d) Any relevant data set names, URLs, DOIs, etc. that an independent researcher would need in order to request your minimal data set. For further information on sharing data that contains sensitive participant information, please see: https://journals.plos.org/plosone/s/data-availability#loc-human-research-participant-data-and-other-sensitive-data If there are ethical, legal, or third-party restrictions upon your dataset, you must provide all of the following details (https://journals.plos.org/plosone/s/data-availability#loc-acceptable-data-access-restrictions):a) A complete description of the datasetb) The nature of the restrictions upon the data (ethical, legal, or owned by a third party) and the reasoning behind themc) The full name of the body imposing the restrictions upon your dataset (ethics committee, institution, data access committee, etc)d) If the data are owned by a third party, confirmation of whether the authors received any special privileges in accessing the data that other researchers would not havee) Direct, non-author contact information (preferably email) for the body imposing the restrictions upon the data, to which data access requests can be sent?

Reviewers' comments:

Reviewer's Responses to Questions

**Comments to the Author**

1. Is the manuscript technically sound, and do the data support the conclusions?

Reviewer #1: Yes

Reviewer #2: Yes

2. Has the statistical analysis been performed appropriately and rigorously?

Reviewer #1: Yes

Reviewer #2: I Don't Know

3. Have the authors made all data underlying the findings in their manuscript fully available?

Reviewer #1: Yes

Reviewer #2: Yes

4. Is the manuscript presented in an intelligible fashion and written in standard English?

Reviewer #1: Yes

Reviewer #2: Yes

**Reviewer #1:**  The manuscript makes a meaningful contribution and demonstrates methodological rigor and clarity. Addressing the minor revisions would improve clarity and impact, especially regarding educational practice and generalizability.

**Reviewer #2:**  The manuscript is well written, novel, and relevant for nursing education. The response rate (89%) is commendable.

Some minor points:

1. There are some ambiguity in terms of "knowledge" and "ability". The manuscript can benefit from clarifying the conceptual difference between "knowledge" and "ability" self-efficacy. E.g., knowledge = “I know what to advise,” ability = “I can conduct a lifestyle counselling session confidently”. Something along that line.

2. Since this study is done in one institution, what the authors would expect for other nursing students, since we have differences in curricula across institutions and countries?

3. In the discussion, perhaps worth highlighting "educational implications" more concretely. Readers would be interested to learn what specific training interventions or curriculum changes would the authors recommend based on these findings?

**Do you want your identity to be public for this peer review?** For information about this choice, including consent withdrawal, please see our Privacy Policy

Reviewer #1: No

Reviewer #2: No

---

## [Author Response · Author response to Decision Letter 1]

20 Jul 2025

Please see the submitted "Response to Reviewers" file, for a detailed point-by-point response to each comment.

---

## [Editor Report · Decision Letter 1]

31 Jul 2025

Nursing Students’ Self-Efficacy in Lifestyle Counselling: Associations with Learning Methods

PONE-D-25-19531R1

Dear Dr. Alenius,

We’re pleased to inform you that your manuscript has been judged scientifically suitable for publication and will be formally accepted for publication once it meets all outstanding technical requirements.

Kind regards,

Pei Boon Ooi, Ph.D.

Academic Editor

PLOS ONE

Additional Editor Comments (optional):

All has been addressed well and accurately- thank you.
---

## [Editor Report · Acceptance letter]

PONE-D-25-19531R1

PLOS ONE

Dear Dr. Alenius,

I'm pleased to inform you that your manuscript has been deemed suitable for publication in PLOS ONE. Congratulations! Your manuscript is now being handed over to our production team.

Kind regards,

on behalf of

Dr. Pei Boon Ooi

Academic Editor

PLOS ONE